# Pigs as Models to Test Cardiovascular Devices

**DOI:** 10.3390/biomedicines12061245

**Published:** 2024-06-03

**Authors:** Yanina L. Rusakova, Denis S. Grankin, Kseniya S. Podolskaya, Irina Yu. Zhuravleva

**Affiliations:** E. Meshalkin National Medical Research Center, Ministry of Health of Russian Federation, 15 Rechkunovskaya St., Novosibirsk 630055, Russia; d_grankin@meshalkin.ru (D.S.G.); podolskaya_k@meshalkin.ru (K.S.P.); zhuravleva_i@meshalkin.ru (I.Y.Z.)

**Keywords:** laboratory mini pigs, transesophageal echocardiography, angiography, aortic morphometry

## Abstract

Pigs as laboratory animals are used in preclinical studies aimed at developing medical devices for cardiac surgery. The anatomy of the cardiovascular system of these animals has been well studied and acknowledged as suitable for use and the testing of new cardiovascular devices developed for humans. However, there are no morphometric characteristics of the aortic root and thoraco-abdominal part of porcine aorta. This can lead to difficulties in experimental surgery and even result in the death of experimental animals due to the mismatch in the size of the implantable devices. Thus, such information is essential to enhance the efficiency of surgical technologies used for eliminating aortic pathologies in their various sections. The purpose of our research is to study the anatomy of the aorta in mini pigs and to assess whether the size, age, and sex of the animals affect the size of the main structures in their aortas. In addition, we attempted to compare the results obtained by transesophageal echocardiography (TEE) and angiography. We studied 28 laboratory mini pigs, dividing them into three groups by body weight (40–70 kg, 71–90 kg, and 90 kg). We did not find any relationship between the external somatometric characteristics of the animals and the size of their aortas. Animals have individual anatomical variability in their cardiovascular systems, which means that they need to be examined in terms of preoperative planning by any available method—echocardiography, angiography, or multispiral computed tomography (CT).

## 1. Introduction

One of the most promising areas in contemporary cardiac surgery is the development of new methods for eliminating pathologies in the aortic valve and thoracoabdominal aorta [1,2,3].

In preclinical studies of new prosthetic devices designed for cardiac valves and thoracic aortas, pigs are the most commonly used models [4,5,6]. The stiffness of the aortic tissue in a young pig is similar to that of the aorta in a healthy person under 60 years of age, particularly in the ascending aorta [7]. The anatomical structure of the aorta in pigs is similar to that in humans, although there are some differences (Figure 1).

At the branching point, the brachiocephalic trunk and the left subclavian artery branch off the aortic arch. In pigs, the common carotid artery with its branches arises from the brachiocephalic trunk. In humans, the most common anatomical variation is when the aortic arch branches into the brachiocephalic trunk, the left common carotid and left subclavian arteries. In humans, this anatomical variant of the arch, similar to that found in pigs, occurs only in 8–30% of individuals [8].

When planning preclinical trials for implantable cardiovascular devices, it is essential to understand not only the general anatomy of the animal, but also the dimensions and geometric characteristics of its aorta. A carefully selected animal model will allow the successful implantation of the device under testing. Moreover, it will help researchers to avoid fatal complications and deaths of experimental animals caused by the mismatch in the sizes of the anatomical structures and medical devices. Multispiral computed tomography is considered to be the most advantageous method for making accurate measurements of the heart and its structures [9,10]. However, in daily experimental practice, it is preferable to use the simplest and quickest method to evaluate the anatomy of the target area. In this case, echocardiography (TEE or transthoracic) or angiography may be useful. Echocardiography is a simple and the most non-invasive technique, but it remains unclear whether the data obtained from this method are comparable to the data from X-rays, which are considered more accurate.

Another important issue under discussion is whether the indicators of mass-growth in adult animals correlate with the size of their cardiac structures. These correlations have been established for young growing animals and humans [11,12,13]. The authors studying the relationships between these parameters conducted experiments on pigs aged between 0 and 3–4 years old. This makes correlations between the size of internal organs and height–weight indicators apparent due to intensive growth during their early years [11,14,15]. However, the growth of internal organs stops when the animal reaches sexual maturity. Therefore, the question we need to address is whether a larger size of cardiovascular structures correlates with a larger body weight.

In different countries, researchers use a variety of laboratory pigs (about 30 different breeding groups) [15]. Hence, for various breeds, different variations in the relationship between height, weight, and the size of cardiac structures can be observed. Consequently, these ratios should be tested prior to the start of a series of experiments to ensure the accuracy of the results and to increase the efficiency of the experiments.

We mainly conduct experiments on mini pigs bred in the nursery at the Institute of Cytology and Genetics SB RAS. Therefore, this study was also done on this particular breed of animals. This breed was developed based on Svetlogorsk mini pigs (54%), large white breed (18%), Landrace (8%), and Vietnamese breed (20%) pigs. Breeding these laboratory animals is possible on a pig farm, while Svetlogorsk mini pigs can only be kept in a vivarium. This breed has a strong constitution and a harmonious build, good reproductive capacity, and it is able to adapt quickly. The average weight of these mini pigs is 110–120 kg, but the main herd maintains a weight range of 50–80 kg. The growth period for these animals is different. Wild boars take 6.6 years to reach their full size, while sows take 3 years [16]. Mini pigs, on the other hand, reach their full size by the age of two. The age-related growth and weight dynamics of mini pigs from this breed are described in detail in the available literature. This includes the growth dynamics of certain internal organs in piglets during the postnatal period [15,16]. However, the main morphometric parameters of the hearts and major vessels in these pigs have not yet been described.

Therefore, the aim of our study is to describe the anatomy of the aorta in mini pigs, to investigate the relationship between the size and age of the animals and the size of the main structures in the aorta, and to determine the usefulness of TEE and angiography in preparing the animal for the experiment.

## 2. Materials and Methods

### 2.1. Study Design

The animals used in the experiment were treated in accordance with the standards set by the European Convention for the Protection of Vertebrates Used for Experiments and Other Scientific Purposes (Strasbourg, 1986).

The study was performed on 28 laboratory mini pigs at the Institute of Cytology and Genetics SB RAS. According to the weight at the time of the study, three groups were identified: 40–70 kg, 71–90 kg, and more than 91 kg.

The animals were taken from the nursery in the Federal Research Center Institute of Cytology and Genetics, Siberian Branch of the Russian Academy of Sciences, Novosibirsk, Russia.

All experimental pigs were first inspected by a vet. Every animal was healthy. Then, the animals were anesthetized. They were weighed, and the body length and the chest volume behind the shoulder blades were measured. Transesophageal echocardiography was performed. Following the echocardiogram, the animals were transferred to the operating room in which angiography was performed. All measurements were carried out by one operator. Each animal was examined once.

### 2.2. Pre-Experimental Manipulations and Anesthesia

The animals were not fed 12 h before the manipulations, but they had free access to water.

For the premedication, Zoletil-100 (Virbac Sante Animale, Carros, France) was administered at 6 mg/kg intramuscularly.

An 18G or 20G peripheral catheter was inserted into the marginal vein of the ear. Animals were measured (weight, chest volume, body length). Then, transesophageal echocardiography was performed. To maintain anesthesia at Stage 1 (measurement) and Stage 2 (TEE), animals were given a bolus of Zoletil-100 at 2 mg/kg and propofol 10 mg/mL (the dose was adjusted according to body weight and depth of introductory anesthesia, on average 2–2.5 mg/kg). For subsequent angiography, the animals were taken to the operating room. Orotracheal intubation was performed in the animal’s “lying on its back” position. Anesthesia was maintained using sevoflurane (3–4 vols.%) with supplemental administration of Zoletil if necessary.

Mini pigs under anesthesia were carefully monitored. Before any incision, their reaction to potentially harmful stimuli, such as a pinching of the tail or ear, was tested at least every 15 min. Throughout the entire process, breathing, the color of the mucous membranes, and the animal’s reaction to manipulation were kept under careful control. Rectal temperature, heart rate, and oxygen saturation were monitored with electronic equipment.

At the end of the manipulations, mini pigs recovering from anesthesia were examined at least every 15 min while they recovered from anesthesia until they regained their consciousness. Once the animals had fully recovered (they were able to hold the position of the sternum and could move around their enclosure), they were returned to their animal housing.

### 2.3. Animal Body Weight Measuring

The age of the animals was determined in accordance with their enclosed documents. Twenty-eight pigs were measured according to the study plan. Weighing the animals was performed on scales designed for weighing animals, VSP4-150 ZHSO VSP4-150 (Neva Scales, Russia manufacturer Nevsky Scales, St. Petersburg, Russia, accuracy class according to GOST OML R76-1-2011: III (medium)).

The volume of the chest behind the shoulder blades was measured with a centimeter tape. The length of the trunk was fixed with a measuring stick (cattle height hip measuring stick) (manufactured by JSC Vetzootechnika Vetzooequipment, Russia) from the occipital protuberance to the tail base.

### 2.4. Ultrasound Study

Echocardiography was performed in accordance with the methods especially designed for animals [17,18] on a Philips CX-50 ultrasound machine (Revision 3.1.2) with an X7-2t transesophageal sensor. The structures of the aortas were evaluated from the fibrous ring of the aortic valve to the ascending part of the aortic arch. The structures of the aorta below were either not visible at all or were difficult to distinguish by echocardiography. Therefore, the size of the aorta below the distal arch was estimated by angiography. The fibrous ring of the aorta was measured at its maximum ring size during systole, with the cursor placed from one inner edge to the inner edge where the flaps attach (Figure 2).

The sinus of Valsalva and the sinotubular junction were measured during diastole from the outer contour to the outer contour (Figure 3).

The height of the aortic root was measured from the fibrous ring to the line of the sinotubular junction.

### 2.5. Angiography Examination

Although angiography is not considered to be the gold standard for evaluating the morphometric parameters of the aorta, some researchers assume the results obtained with angiography to be more accurate than those obtained from computed tomography or magnetic resonance angiography [19,20]. With angiography, the ascending aorta, the proximal part of the aortic arch and the main vessels arising from it, as well as the distal part of the aortic arch, isthmus, thoracic, and abdominal arteries can be clearly visualized. The diameter of these vessels and their length can be measured. In our study, echocardiography was first performed, and then the animals were taken to the operating room to perform angiography. The animals were fixed in a position on their backs. To carry out angiography of the aorta, a 7Fr section introducer was inserted into the left femoral artery. Supportive infusion therapy was administered into the marginal vein of the ear or into the jugular vein through an introducer. Before the use of endovascular instruments, the animal was injected with heparin at a dosage of 100 units per kg of body weight. A diagnostic Cordis 6F diagnostic catheter, Miami Lakes, FL, USA, was inserted into the area of interest using a diagnostic hydrophilic conductor (0.035″ diameter, 260 cm long, with a J-3 mm tip). The examination began at the aortic root. Then, the diagnostic catheter was moved gradually, with a series of straight and oblique images being taken. The diameter of the ascending and descending aortic arch and the main vessels arising from it—the brachiocephalic trunk and the left subclavian artery, the thoracic aorta at the diaphragm level, in the abdominal department at the bifurcation level—were estimated. For angiography, the C-Arc OEC 9900 Elit (General Electric, Boston, MA, USA) was used, and a series of images were taken using the contrast agent Omnipak (LLC Scientific and Technological Pharmaceutical Company POLISAN, Saint Petersburg, Russia). Omnipack was administered manually using a 10 mL syringe at a rate of 5 mL/s. Angiography was performed using an anterior–posterior projection. At the end of the examination, the femoral artery was ligated with the absorbable suture material Safil 0 (3.5) (Safil, B Braun, Melsungen, Germany); the wound was stitched layer by layer with the same suture material. The obtained images were processed using the program RadiAnt (DICOM Viewer 4.6.9 (64-bit) Medixant, Poznan, Poland).

The type of aortic root was determined based on the procedure that was proposed in 2022 for determining the type of aortic root in humans [21]. The aortic root was thought to be a part of the ascending aorta from the fibrous ring of the aortic artery to the plane of the sinotubular junction. Taking into account the ratio of the height of the aortic root and the diameter of the fibrous ring, the authors identified three variants of the structure of the aortic root: A, B, and C. Type A is characterized by a large value of the height of the of the Valsalva sinuses in relation to the diameter of the fibrous ring (K > 1.05). In the aortic root of Type B, the height of the sinuses of the Valsalva is close in magnitude to the diameter of the fibrous ring (0.95 ≤ K ≤ 1.05). In the root of Type C aorta, the height of the sinuses of the Valsalva is less than the diameter of the fibrous ring (K ≤ 0.95). The coefficient is calculated by the formula K= h/D, where h is the distance from the fibrous ring to the sinotubular joint (mm) and D is the average diameter of the fibrous ring of this animal (mm). The K coefficient was calculated for each mini pig.

### 2.6. Statistical Analysis

Quantitative data were processed using Dell Statistica 13.0 (Dell Software Inc., Round Rock, TX, USA). As most groups’ distributions were not typical, non-parametric statistics were used. Quantitative data are reported as medians and interquartile ranges (25–75%) (IQRs). The Kruskal–Wallis (K–W) test was used to compare three or more groups. The Spearman correlation was used to identify the relationships between the values. The level of significance was set to *p* < 0.05.

## 3. Results

### 3.1. The Results of Measuring the Animals’ Bodies

Table 1 shows that the average weight of mini pigs reaches 65 kg by the age of one year, and about 80 kg by the age of two. Their growth does not stop then. At the age of three years, laboratory mini pigs bred by ICiG SB RAS can weigh about 100 kg. At the same time, the ratio of their chest volume behind the shoulder blades to their body length (the index of animal fatness) was within the range of 0.87–1.07 (transitional or eirisomal constitutional type) in all three groups, regardless of age (Table 1).

There seems to be a strong direct positive correlation between somatometric parameters, as confirmed by Spearman’s nonparametric method (body length and breast volume (*p* = 0.82, at *p* < 0.05), body length and weight (*p* = 0.78, at *p* < 0.05) and body weight and breast volume (*p* = 0.87, at *p* < 0.05)).

### 3.2. The Results of the Ultrasound Study

The results of the TEE examination are presented in Table 2.

All of the animals’ structures, such as the fibrous ring of the aortic valve, the sinus of Valsalva, and the sinotubular junction, were clearly identified by transesophageal echocardiography (Figure 4).

Visualization of the ascending aortic arch, the proximal part of the arch, the subclavian artery, and the brachiocephalic trunk was challenging or impossible.

Given the data obtained by this method, the diameter of the fibrous ring of the aortic valve in Group I pigs averaged 20.7 (19.6/23.7) mm, in Group II pigs—22.8 (16.4/24.0) mm, and in Group III pigs—25.0 (23.5/28.8) mm. At the level of the sinuses of Valsalva, the diameter of the aorta was 31.0 (28.9/33.0) mm in Group I, 29.8 (24.1/36.0) mm in Group II, and 34.8 (30.9/40.0) mm in Group III. At the level of the sino-tubular junction, the median values of the aortic diameter in Group I were 23 mm, in Group II—24.7 mm, and in Group III—27.9 mm (Table 2). Table 2 shows that the median values of the diameters of aortic structures rise with the increase of animal weight and age. However, the differences between them are invalid (the *p* value according to the Kraskel–Wallis criterion was more than 0.999, while the statistically significant *p* value should be less than 0.017 when these three groups are compared).

The significant individual variability of indicators in each group of animals is well demonstrated in Figure 5.

It was found that there was a strong positive relationship between certain morphological structures of the aortic root. Specifically, between the diameter of the fibrous ring of the aorta and the sinuses of the Valsalva, the Spearman correlation was *p* = 0.73 (at *p* < 0.05); between the diameter of the fibrous ring of the aorta and the sinotubular junction, the correlation was *p* = 0.80 (at *p* < 0.05); the correlation between the sizes of the sinuses of Valsalva and the ascending aorta was *p* = 0.80 (at *p* < 0.05); and the correlation between the sizes of the sinotubular junction and the ascending aorta was *p* = 0.72 (at *p* < 0.05).

### 3.3. The Results of the Angiography Examination

Angiography allows for the visualization of both the aortic root and its thoracic and abdominal sections at any level prior to bifurcation (Figure 6).

The results of the angiography examination are given in Table 3.

When comparing the echocardiogram data with the results obtained from angiography (Table 2 and Table 3), it can be seen that the differences between them are insignificant and, in general, do not exceed 1 mm. The Spearman coefficient between the data obtained by these methods was *p* = 0.84–0.98 at *p* < 0.05 (Figure 7).

Angiography allowed us to observe a gradual narrowing of the diameter of the porcine aorta from the distal part of the aortic arch to the bifurcation in the abdominal area. Moreover, the most significant reduction in diameter was seen in the Group I animals. Notably, in the isthmus zone, the diameter of the aorta decreased relative to the distal part of the arch in Group I animals by 18%, in Group II by 7%, and in Group III by 0.6%. In the diaphragm area, the diameter of the aorta was smaller in relation to the distal part of its arch in each group (Group I, Group II, and Group III) by 45%, 23.5%, and 13.6%, respectively. In the bifurcation area, the reduction in the diameter of the aorta relative to the distal part of its arch in Groups I, II, and III was 51%, 52%, and 36%, respectively (Figure 8).

It was found that there was a moderate positive correlation between certain external parameters of the animals and the diameter of the abdominal aorta. Namely, there was a correlation between the volume of the breast and the diameter of the aorta at the diaphragm level (*p* = 0.54, at *p* < 0.05) and between the weight of the pigs and the diameter of the aorta at the diaphragm point (*p* = 0.46, at *p* < 0.05). Spearman’s rank correlation was *p* = 0.5, at *p* < 0.05 and it involved body length and aortic diameter at the bifurcation level, breast volume and aortic diameter at the bifurcation level, weight, and aortic diameter at the bifurcation level.

## 4. Discussion

To visualize the ascending aorta and the aortic root in animals with a smaller body weight, a parasternal position along the long axis of the left ventricle can be used [22,23]. However, due to the poor visualization provided by transthoracic echocardiography, it can be challenging to obtain accurate results when pigs weighing more than 60 kg are examined even when using a specialized echocardiography table for animals. In our study, we used a transesophageal sensor. It allowed us to evaluate the geometry of the aortic root from the mid-portion of the esophagus. When it is necessary to make accurate measurements of these parameters in pigs, as well as the diameters of the thoracic and abdominal aortas at any level before bifurcation, angiography can be performed.

According to the published data, in humans, the diameter of the sinotubular junction remains larger relative to the fibrous ring of the aorta by 10–20% [24]. The diameter of the sinus of Valsalva in humans is 19% larger than that of the ascending aorta, and the diameter of the descending aorta is 24% smaller on average [25]. We have not observed similar patterns in pigs. Our results showed that the diameter of the sinotubular junction was 8–11% larger than that of the fibrous ring of the aorta and the diameter of the sinuses of Valsalva exceeded the diameter of the ascending aorta in pigs weighing up to 70 kg by 29%, and in pigs weighing more than 70 kg by 48%. The diameter of the ascending aorta was 11–26% larger than that of the descending one.

However, unlike other researchers, we have not found a clear relationship between the body length, chest volume, and adult mini pig weight and the size of the animals’ thoracic aorta or the structures of its root. For example, in large Swiss white pigs, the length of the ascending aorta correlates with body weight [14], or in miniature inbred pigs, the body length correlates with the aortic ring and the diameter of the aortic root according to Allan’s study [11].

It can be said that the aortas of larger pigs are not always larger than those of smaller animals. The lack of correlation between the weight of pigs and the size of the thoracic/abdominal aorta is supported by the data presented in the manual for anesthesia, surgery, and experimental work with laboratory pigs [26]. It was suggested there that Yucatan pigs weighing 29 kg had a diameter of 12 mm in their thoracic aortas and 6 mm in the postrenal sections/segments of their abdominal area. Yucatan pigs weighing 109 kg had diameters of 11 mm in their thoraxes and 6 mm in their abdomens. The manual for other pig breeds also notes that there is no correlation between the weight of the animals and the diameter of their aortas (Figure 9).

Our results are consistent with the data obtained at the Quebec Heart and Lung Institute, QC, Canada [27]. Georges G et al. noted that the reduction in the diameter of the aortas in pigs in the abdominal region compared to the thoracic was from 19% to 48%. In our mini pigs, the diameters of the thoracic and abdominal aortas were slightly smaller than those described by Georges G. It was stated there that in animals weighing 40–59 kg, the diameter of the thoracic aorta was on average 20.0 ± 1.7 mm, and the abdominal 10.5 ± 3.8 mm; in pigs weighing 60–79 kg, the same diameters were 18.6 ± 2.0 mm and 15.1 ± 2.2 mm, respectively; pigs weighing more than 80 kg had thoracic aortas with a diameter of 21.4 ± 2.2 mm and abdominal aortas of 11.0 mm. However, their study did not show at which points the measurements of the thoracic and abdominal aortas were made.

Regarding the structural variant of the aortic root, approximately half of the animals studied (46.43%) had a height of the aortic root that was less than that of the fibrous ring, which is known as the “C” root type (K ≤ 0.95). The second half (46.43%) showed the type “A” root (the height of the sinuses being greater than the fibrous ring). In 7.14% of the cases (in two animals out of twenty-eight animals) it was found to be the type “B” root (the height of the sinuses being approximately equal to the diameter of the fibrous ring in the aorta). Moreover, type “A” was found mainly in pigs weighing 71–90 kg, and type “C” in animals weighing 40–60 and more than 90 kg. In a human, the aortic type C root can be found in 98% of cases, and type B in 1.6% of cases; type A almost never occurs (0.4%) (Kobelev E. et al. [21]) (Figure 10).

Special attention should be given to certain types of aortic root in experimental animals when designing new models of transcatheter aortic valves for humans. Therefore, it is suggested that preclinical models should be as closely aligned with the final product as possible.

Thus, taking into account the individual variability in the aortic root structure among pigs, it is difficult to overstate the importance of preoperative echocardiography in experimental cardiac surgery [28,29,30]. When measuring the parameters of the aortic root, echocardiography is the most convenient, quickly feasible method as it is much less invasive compared to angiography. However, angiography provides excellent visualization of the thoracic and abdominal parts of the aorta.

## 5. Conclusions

We have conducted a morphometric analysis of the aorta in laboratory mini pigs. The results obtained have demonstrated that there is an individual variability in the geometry of the aortic root that is not related to the somatometric characteristics of the animals. This indicates the need for the careful selection of laboratory animals for use as models. In addition, high-tech and innovative methods for preoperative planning, such as computed tomography, can be implemented in experimental research. If the latter is not an option, echocardiography and angiography can help us to evaluate the anatomical structures of potential models preliminarily. Unfortunately, there are currently no pig breeds that have standardized aortic sizes and geometries. The anatomy of each animal is unique and does not depend on their weight. Our research was conducted on 28 mini pigs. Therefore, the number of animals under study should be increased significantly in groups, and this will allow us to properly identify a correlation between the somatometric characteristics of the animals and the diameter of their major vessels and the valve apparatus of the heart.

To sum up, we hope that the results of our research will help researchers select an appropriate model for preclinical trials and contribute to the further development of aortic devices in order to avoid mismatch inconsistencies in the sizes of anatomical structures and medical devices.

## Figures and Tables

**Figure 1 biomedicines-12-01245-f001:**
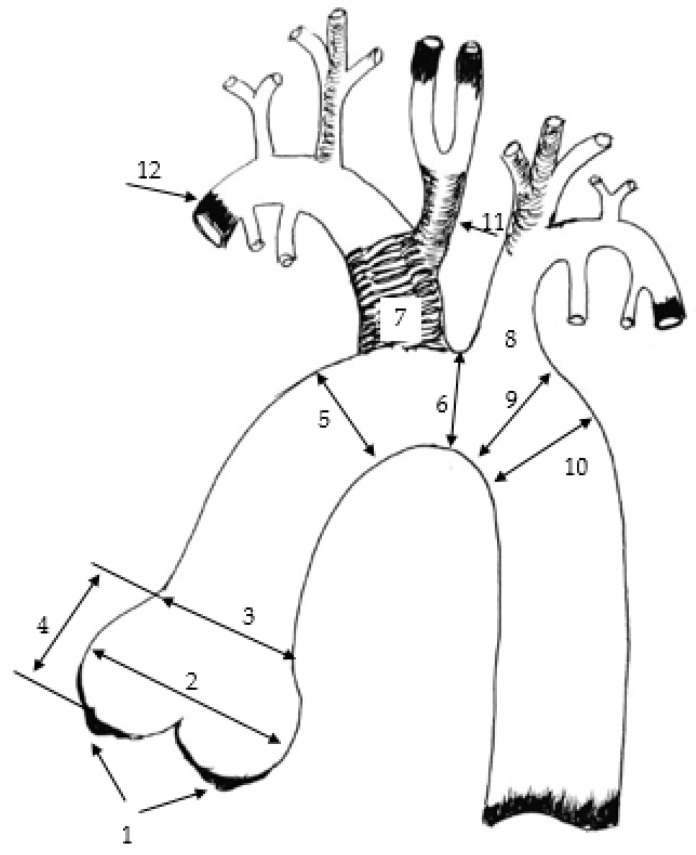
The scheme of branching of the main arteries in pigs. 1—aortic valve annulus fibrosus (AoVAnF), 2—sinus of Valsalva (SVal), 3—sinotubular junction (StJ), 4—aortic root height (Ao root height), 5—ascending aorta (Asc Ao), 6—proximal portion of the aortic arch (ProxAo arch), 7—brachiocephalic trunk (BrachTr), 8—left subclavian artery (LSubclAa), 9—distal portion of the aortic arch (DistAo arch), 10—aortic isthmus (Ao Ist), 11—common carotid artery, 12—right subclavian artery.

**Figure 2 biomedicines-12-01245-f002:**
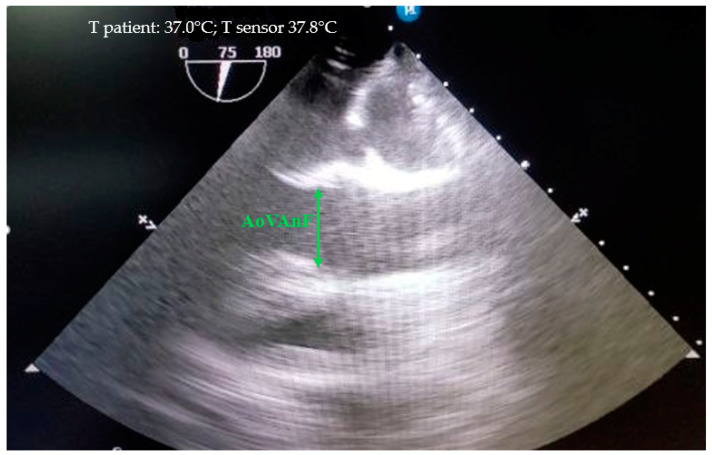
Measurement of the fibrous aortic ring (AoVAnF) in the mini pigs from the “inner edge to the inner edge” at the site of flap attachment, where the ring reaches its maximum size.

**Figure 3 biomedicines-12-01245-f003:**
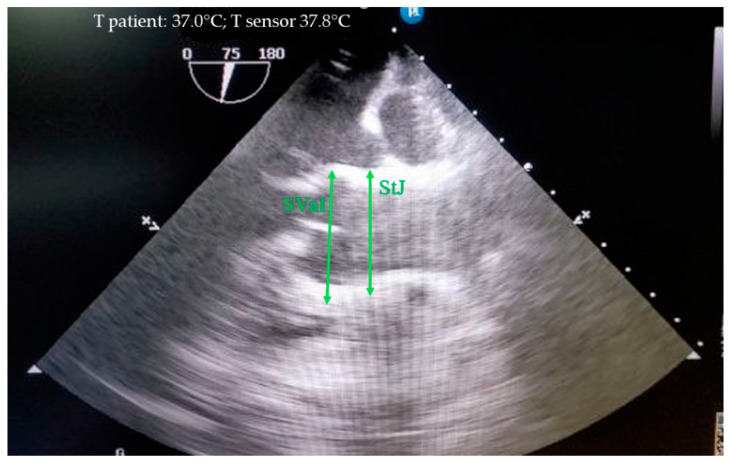
Measurement of the sinus of Valsalva (SVal) and sinotubular junction (StJ) in mini pigs from the outer contour to the outer contour during diastole.

**Figure 4 biomedicines-12-01245-f004:**
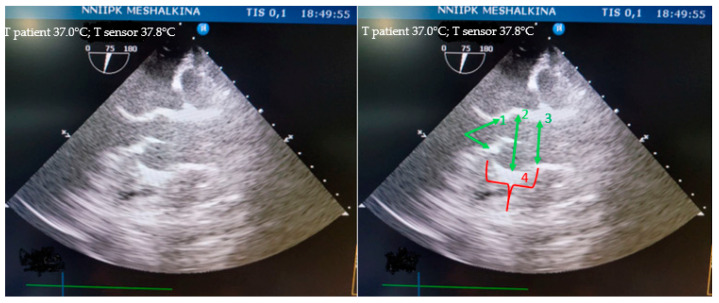
Visualization of the aortic root in mini pigs on TEE: 1—fibrous ring of the aortic valve; 2—sinus of Valsalva; 3—sinotubular junction; 4—height of the aortic root.

**Figure 5 biomedicines-12-01245-f005:**
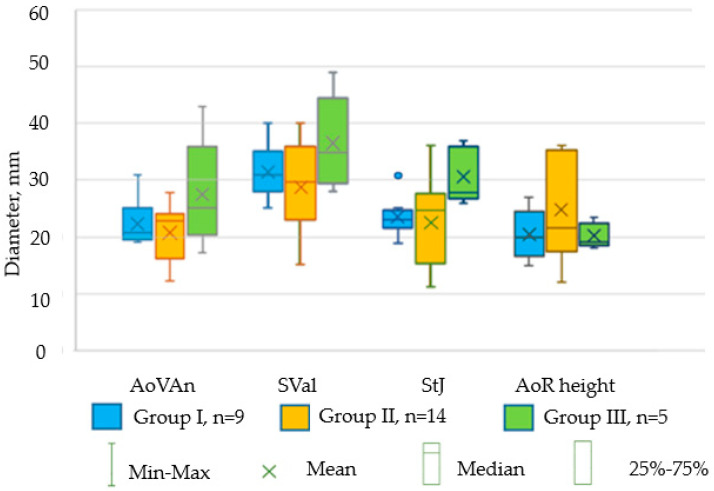
The main parameters of the aortic root in mini pigs in groups of 40–70 kg (Group I, n = 9), 71–90 kg (Group II, n = 14), and more than 90 kg (Group III, n = 5)). AoVAn—diameter of the fibrous ring of the aorta according to TEE; SVal—sinus Valsalva according to TEE; StJ—sinotubular junction according to TEE; AoR height—height of the aortic root according to TEE. Along the *x*-axis, there are the main structures of the aortic root; the *y*-axis shows the size of the diameter in mm. No significant differences in the size of aortic valve annulus fibrosus, sinus of Valsalva, sinotubular junction, and aortic root height were found in all three groups (*p* > 0.999 in the K–W test).

**Figure 6 biomedicines-12-01245-f006:**
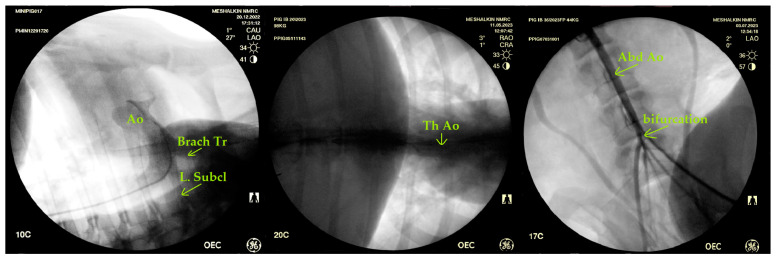
Visualization of the aorta in its various sections by angiography: Ao—aorta; Brach. Tr.—brachiocephalic trunk; L. Subcl.—left subclavian artery; bifurcation- bifurcation; Abd.Ao—abdominal aorta; Tr Ao—thoracic aorta.

**Figure 7 biomedicines-12-01245-f007:**
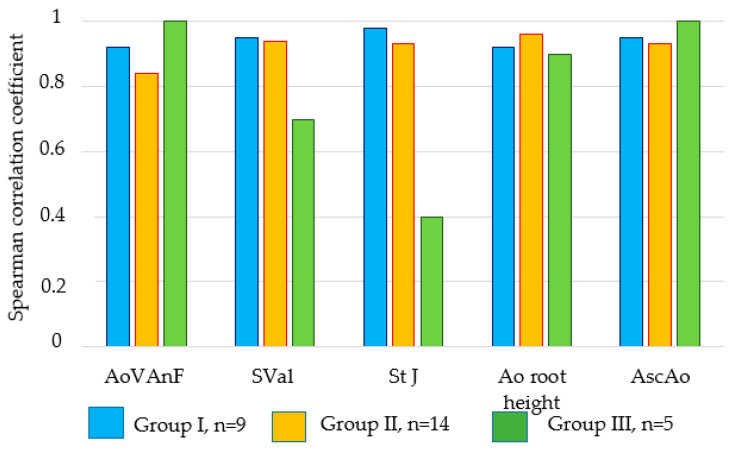
Spearman correlation’s rank order between the results obtained by echocardiography and angiography; the noted correlations are significant at *p* < 0.05. Along the *x*-axis, the main structures of the aortic root are given; the *y*-axis shows the value of the Spearman correlation coefficient.

**Figure 8 biomedicines-12-01245-f008:**
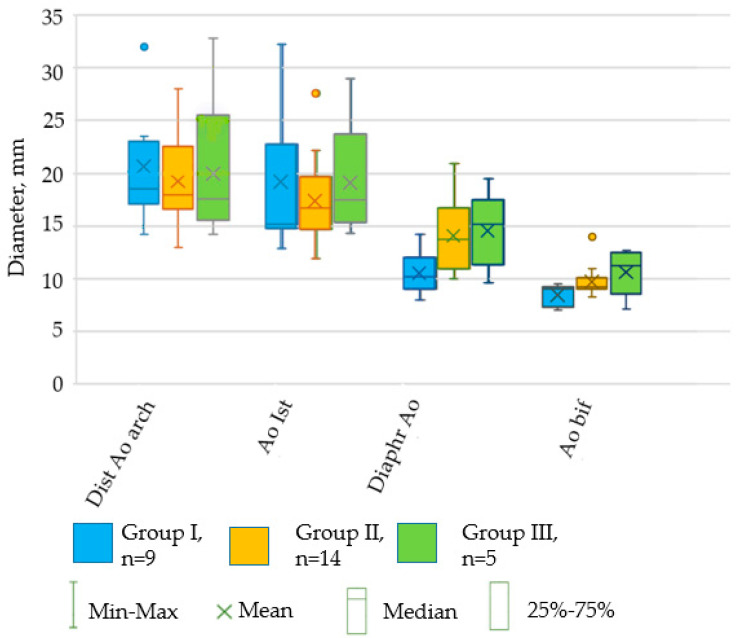
The change in the diameter of the aorta (mm) from the distal part of the aortic arch to the bifurcation in the abdominal region. Dist Ao arch—distal portion of aortic arch; Ao Ist—aortic isthmus; Diaphr Ao—diaphfragmatic part of the aorta; Ao bif—aortic bifurcation. There were no significant differences between the groups in the size of the aortic diameter at these measurement points (*p* = 1 in the K–W test). The measurement points of the aorta are plotted along the *x*-axis; the *y*-axis shows the size of the diameter in mm.

**Figure 9 biomedicines-12-01245-f009:**
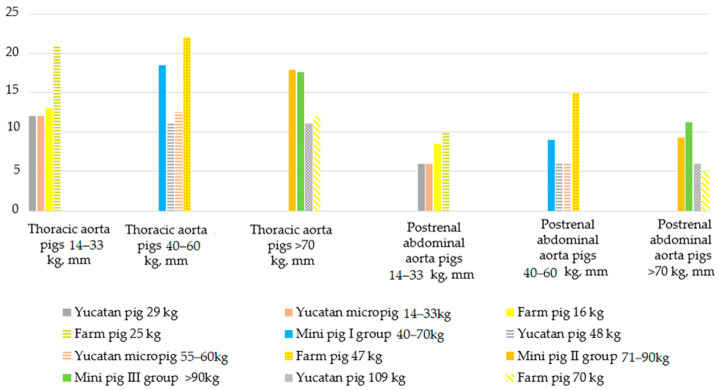
The change in the diameter of the thoracic and abdominal aortas in mini pigs of different weights compared to pigs of other breeds (Swindle M.M. et al., 2016 [26]). The measurement points of the aortas are plotted along the *x*-axis; the *y*-axis shows the size of the diameter in mm.

**Figure 10 biomedicines-12-01245-f010:**
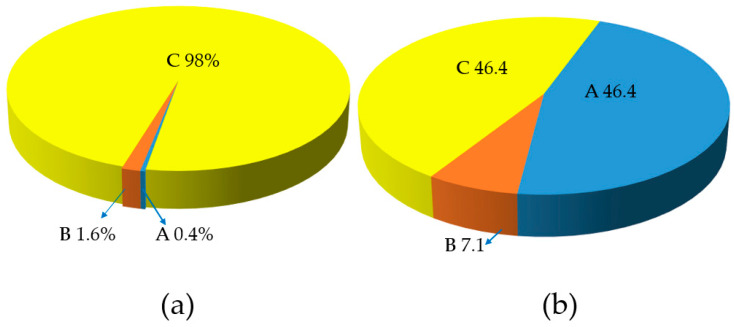
The occurrence of the different types of aorta in humans and in mini pigs: (**a**) percentage of occurrence in humans (n = 251); (**b**) percentage of occurrence in pigs (n = 28).

**Table 1 biomedicines-12-01245-t001:** Parameters of the animals’ bodies in the tested groups, *p* ≤ 0.017 *.

Parameter	I Group, n = 9 (40–70 kg) Median (Q25%/Q75%)	II Group, n = 14 (71–90 kg) Median (Q25%/Q75%)	III Group, n = 5 (>90 kg) Median (Q25%/Q75%)
Male/Female, n, (%)	4/5	3/11	0/5
	(44.4/55.6%)	(21.4/78.6%)	(0/100%)
Age, month	12.0 (8/18)	24.0 (24/36)	36.0 (32/36)
Weight, kg	65.0 (53/68)	76.50 (76/86)	98.0 (93/101)
Body length, cm	94.0 (90/95)	99.0 (97/105)	111.0 (110/118)
Breast volume, cm	90.0 (86/95)	101.5 (98/109)	114.0 (114/120)

* There were no significant differences in age, weight, body length, and breast volume in all three groups (*p* > 0.05 in the K–W test).

**Table 2 biomedicines-12-01245-t002:** Aortic dimensions in miniature pigs taken by using transesophageal echocardiography, n = 28, *p* ≤ 0.017 *.

Parameter	I Group, n = 9 (40–70 kg) Median (Q25%/Q75%)	II Group, n = 14 (71–90 kg) Median (Q25%/Q75%)	III Group, n = 5 (>90 kg) Median (Q25%/Q75%)
Aortic valve annulus fibrosus, mm	20.70 (19.6/23.7)	22.8 (16.4/24.0)	25.0 (23.5/28.8)
Sinus of Valsalva, mm	31.0 (28.9/33.0)	29.8 (24.1/36.0)	34.8 (30.9/40.0)
Sinotubular junction, mm	23.0 (22.1/24.5)	24.7 (15.4/27.2)	27.9 (27.5/35.0)
Aortic root height, mm	19.9 (17.7/22.6)	21.6 (17.5/35.0)	19.0 (18.9/21.2)
Ascending aorta, mm	24.0 (22.2/25.4)	20.1 (16.0/25.3)	23.7 (19.3/28.0)

* No significant differences in the size of aortic valve annulus fibrosus, sinus of Valsalva, sinotubular junction, aortic root height, and ascending aorta were found in all three groups (*p* > 0.999 in the K–W test).

**Table 3 biomedicines-12-01245-t003:** Aortic dimensions in miniature pigs taken using angiography, *p* ≤ 0.017 *.

Parameter	I Group (n = 9) (40–70 kg) Median (Q25%/Q75%)	II Group (n = 14) (71–90 kg) Median (Q25%/Q75%)	III Group (n = 5) (>90 kg) Median (Q25%/Q75%)
Aortic valve annulus fibrosus, mm	21.1 (20.0/23.5)	22.5 (17.0/23.3)	24.6 (21.2/28.2)
Sinus of Valsalva, mm	31.5 (28.9/32.6)	28.2 (24.3/35.8)	29.6 (27.1/40.3)
Sinotubular junction, mm	23.0 (22.2/24.0)	25.2 (17.0/27.0)	22.9 (20.8/23.6)
Aortic root height, mm	18.5 (16.0/20.9)	21.9 (18.0/35.2)	20.8 (20.2/22.9)
Ascending aorta, mm	23.8 (22.0/32.4)	19.6 (17.5/25.0)	23.4 (19.3/25.0)
Proximal portion of aortic arch, mm	22.4 (19.6/24.0)	20.2 (16.0/25.5)	24.0 (20.8/25.0)
Distal portion of aortic arch, mm	18.5 (17.9/22.5)	17.9 (16.8/22.0)	17.6 (17.0/18.2)
Aortic isthmus, mm	15.2 (15.0/22.4)	16.7 (15.0/19.6)	17.5 (16.5/18.3)
Diaphragmatic part of the aorta, mm	10.2 (9.0/12.0)	13.7 (11.0/16.6)	15.2 (13.0/15.5
Aortic bifurcation, mm	9.0 (7.7/9.0)	9.3 (9.0/10.0)	11.2 (10.0/12.2)
Brachiocephalic trunk, mm	10.0 (9.0/10.0)	10.9 (10.0/11.1)	14.7 (10.2/15.1)
Left subclavian artery, mm	9.0 (8.8/9.0)	9.6 (8.0/10.0)	11.8 (11.2/12.0)

* No significant differences in the size of aortic structures were found in all three groups (*p* > 0.999 in the K–W test).

## Data Availability

The data presented in this study are available within the article.

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
