# Peer review of "Pigs as Models to Test Cardiovascular Devices"

_biomedicines, 2024, doi:10.3390/biomedicines12061245_

Round 1

Reviewer 1 Report

Comments and Suggestions for Authors

I have some major concerns for this manuscript entitled “A pig as a model to test cardiovascular devices” as followed.

1. There are some logical errors in the abstract, such as the second sentence on the one hand, well studied and acknowledged…” and the third sentence on the other hand, there are no morphometric characteristics…”. The relationship of these sentences is adversative, not progressive. Besides, at the end of the abstract, is the conclusion of the study each animal has an individual anatomy? The conclusion should be stated clearly.

2. Some changes are needed to enhance the connection and logic among the paragraphs in the Introduction.

3. In the introduction, “The live weight of 76 these mini pigs reaches 110-120 kg in boars and 70 kg in sows”. However, the weight of some mini-pigs used in the study was more than 91kg. Is this contradictory?

4. How to judge the depth of anesthesia of mini-pigs?

5. Anesthetics have a great influence on the body temperature of mini-pigs. How to rule out the possible influence of body temperature on the cardiovascular system when conducting various measurements? Have insulation measures been taken?

6. In the Discussion, the relationship between this study and the clinical use should be emphasized.

Comments on the Quality of English Language

Must be improved by the native speaker.

Reviewer 2 Report

Comments and Suggestions for Authors

The authors Rusakova et al. studied the anatomy of the aorta in pigs to find common correlations between age, size or morphometric parameters and aortic dimensions in view of research on pre-clinical studies for developing aortic device used later on in humans. To follow their aim, they used 28 mini-pigs, divided into 3 groups according to age. Aortic dimensions were measured at pre-defined levels of the vessel by ultrasound and by angiography. While the goal is a very worthy one, due to the increased need of vascular devices and the need of reference books for animal studies, the manuscript needs a thorough revision. Firstly, it is recommended to restructure each section, as the content of each is very confusing and not to the point. Several important information is missing: As the authors cite that the strain of animals was bred at the institute of Cytology and Genetics SBRAS (lines 69-71) and afterwards give examples of differences towards other commonly used strains, they should state whether this specially bred strain has already been used in aortic device research. In the methodology section, the experimental conditions should be better specified (i.e.: flow rate of contrast agent during angiography, protocol of angiography projections, who performed angiography – same researches as the echocardiography or not, inter-/intra-observer variability and so on). Most of all, the results should be sorted by paragraphs to demonstrate the different sub-questions the authors wished to address. In the current version, the reader gets lost among the multiple parameters without getting any substantial main results from the data. The figures all need improvement, for example by adding labels to the axes, and legends and lists of abbreviations to the figures. Several figures should be re-drawn completely, such as Figure 7 or 8, as their content does not become clear even after thorough study. The discussion lacks the sum up of the results and reference to current knowledge, but loses the reader among new numbers and details.

Comments on the Quality of English Language

It is recommended to perform editing of the English language during revision.

Reviewer 3 Report

Comments and Suggestions for Authors

Rusakova et al reported their work named "A pig as a model to test cardiovascular devices" and concluded, "Each animal has an individual anatomy, so they are to be examined 21 in terms of preoperative planning by any available method - echocardiography, angiography or multispiral computed tomography (MRI).". I have the following comments:

- Please mention that you have 3 groups in the abstract methods.

- Please specify the reason for choosing those 3 groups and the cutoffs in your methods.  Did you try to group them into 2 groups based on the median weight? This can serve as a supplementary/sensitivity analysis.

- The authors mentioned mini-pigs but later they mentioned different types of pigs in Figure 10.

- Please add a reference to that in your figure legend

- Minor language edits are essential eg "Correlation Spearman" should be "Spearman correlation"

- Please add p-value to your table 1.

- Figure 4: Please either delete this figure or transfer it to supplements. Please add a reference to the Figure 1a legend.

- Please acknowledge the small sample size at the end of your discussion.

- Please add conclusion to your abstract and manuscript.  

Comments on the Quality of English Language

- Minor language edits are essential eg "Correlation Spearman" should be "Spearman correlation"

Round 2

Reviewer 1 Report

Comments and Suggestions for Authors

NA

Comments on the Quality of English Language

NA

Reviewer 2 Report

Comments and Suggestions for Authors

During the revision process, the authors Rusakova et al. answered each comment of this reviewer. This led to minimal changes of the manuscript text and of the figure legends, and to the addition of a drawing of the human aortic anatomy (presumably, as the legend is missing). Some of the figures were clearly enhanced by the addition of measurement points and arrows; however, as the figures are not drawn in uniform style, they are still difficult to interprete. Although answered, some of the comments were not transferred into the manuscript, such as the question of who performed the experiements, the number of repetitions, the inter/intra-observer variability etc. The results are still difficult to follow and should be ordered to improve understanding. Altogether, the authors have undertaken some efforts to refine the manuscript. In the current version, the manuscript offers still some room for development.

Comments on the Quality of English Language

Some minor corrections might be required.
